# Understanding Problematic Social Media Use in Adolescents with Attention-Deficit/Hyperactivity Disorder (ADHD): A Narrative Review and Clinical Recommendations

**DOI:** 10.3390/brainsci12121625

**Published:** 2022-11-26

**Authors:** Tycho J. Dekkers, Jorien van Hoorn

**Affiliations:** 1University Medical Center Groningen, Department of Child and Adolescent Psychiatry, University of Groningen, 9723 HE Groningen, The Netherlands; 2Accare Child Study Center, 9713 GZ Groningen, The Netherlands; 3Levvel, Academic Center for Child and Adolescent Psychiatry, 1105 AZ Amsterdam, The Netherlands; 4Amsterdam University Medical Center (AUMC), Department of Child and Adolescent Psychiatry, 1100 DD Amsterdam, The Netherlands; 5Department of Developmental and Educational Psychology, Leiden University, 2311 EZ Leiden, The Netherlands

**Keywords:** Attention-Deficit/Hyperactivity Disorder (ADHD), social problems, online, social media, adolescents, review, brain development, peers, parents, clinical recommendations

## Abstract

Attention-Deficit/Hyperactivity Disorder (ADHD) is consistently associated with a host of social problems, such as victimization and difficulties in maintaining close friendships. These problems are not limited to offline relations but also manifest in the online social world, as previous research shows that ADHD is associated with problematic use of social media. Given the ubiquitous nature of social media, the goal of the current review is to understand why adolescents with ADHD demonstrate more problematic social media use than their typically developing peers. To this end, we provide a narrative review on the evidence for the link between ADHD and social media use, and consequently present an integrative framework, which encompasses neurobiological mechanisms (i.e., imbalance theory of brain development and dual pathway model of ADHD) and social mechanisms, including influences from peers and parents. We conclude that empirical work shows most consistent evidence for the link between problematic social media use and ADHD (symptoms), while intensity of social media use is also associated with several other behaviors and outcomes. Finally, we hypothesize how existing interventions for ADHD may work on the identified mechanisms and provide at-hand clinical recommendations for therapists working with adolescents with ADHD who exhibit problematic social media use.

## 1. Introduction

Attention-Deficit/Hyperactivity Disorder (ADHD) is a highly prevalent psychiatric condition characterized by inattention, hyperactivity and impulsivity [1]. ADHD is consistently associated with a host of social problems, including difficulties to manage peer relations and close friendships [2,3,4]. This is especially problematic during adolescence, the developmental period between childhood and adulthood, when belonging to the peer group becomes highly salient [5]. One prominent aspect of adolescents’ social lives takes place online through social media [6]. A total of 45% of adolescents report being online almost constantly [7], and on average 7.5 h a day is spent on screens, of which a significant portion is dedicated to social media [8].

Previous work has linked ADHD to more frequent and more problematic use of social media [9,10]. The latter is more complex than merely spending many hours a day on social media. In fact, *problematic* social media use can be seen as an enduring preoccupation with and inability to stop using social media, as well as a persistent neglect of one’s health and functioning in several life domains, such as sleep, family, friends and school [11,12]. While social media offers an opportunity for self-expression and connecting to peers, other work has highlighted potential negative consequences for mental health, as the online world can be a platform for bullying, victimization, sexting and risky behavior (reviewed in [13,14,15]). In this narrative review, we summarize recent evidence on the relation between ADHD and social media use, and provide an integrative framework of explanatory mechanisms for social media use from a neurobiological perspective (i.e., imbalance model of brain development, dual systems and reinforcement sensitivity theories of ADHD) as well as social perspective (i.e., peer and parent relationships) [16,17,18,19,20,21,22]. Finally, we identify research gaps and provide clinical recommendations for therapists working with adolescents with ADHD who exhibit problematic social media use.

## 2. ADHD and (Social) Media Use: A Brief Literature Review

Most research on ADHD and media use in recent years has examined either general screen time or digital media use, or zoomed in on specific digital domains such as watching television or internet addiction. We first provide a comprehensive overview of what is known about ADHD and digital media, and then situate studies on ADHD and *social* media within this broader field. Social media can be defined as ever-changing, immersive and continually engaging [23], as they enable people to exchange information, ideas, messages, images and videos [24]. We discuss recent longitudinal studies and meta-analyses, which allow for the most robust conclusions about individual differences and changes over time [25], complemented by empirical work that differentiates between the intensity of social media use, which is not necessarily related to impairment [26], but see [27], and problems related to social media use [11].

Several longitudinal studies from the communication sciences literature appear to suggest a temporal relationship between media use and subsequent ADHD characteristics, although most studies in this field did not include groups with clinical ADHD diagnoses, and potential confounds of this work and the implications have been widely debated. In the domain of television consumption, a Brazilian cohort study with adolescents from the general population found that ADHD symptoms at age 22 were related to television time at age 11, and total screen time at age 18 [28]. Another study has shown that the frequency of watching television at age 14 is linked to attention problems at age 16 after controlling for prior family and cognitive factors [29]. Similarly, a longitudinal cohort study from New Zealand reported a significant association between hours of watching television as a child, and attention problems in adolescence, even after controlling for other relevant variables, such as attention problems in childhood [30]. 

In addition, a large longitudinal study in adolescents without significant ADHD symptoms found that higher engagement in digital media activities (either social or non-social) was associated with more self-reported ADHD symptoms two years later [10]. This study concludes that “modern digital media use could play a role in the development of ADHD symptoms”. Yet, several confounders were proposed by the authors and other researchers (e.g., parenting style, parent media use, family chaos, sleep, peer activities [31,32]) and the diagnostic rigor of the study was criticized [32]. Finally, another recent study illustrated that self-reported social media use problems, but not social media use intensity, predicted self-reported ADHD symptoms over time [33].

Three recent meta-analyses reach less straightforward conclusions. A small relation between ADHD characteristics and media use is demonstrated by one meta-analysis (*r* = 0.12 [34]), while another found a very small association between ADHD characteristics and video gaming (*r* = 0.03 [35]), and a third yielded a moderate link between ADHD and internet addiction in adolescents and young adults (adjusted OR = 2.51 [36]). Causality from these meta-analyses is largely unclear, because transactional and bidirectional relationships are likely. Moreover, Beyens and colleagues [9] note the importance of individual differences, as tentative evidence suggests that male sex and aggression in adolescents, as well as less responsive parenting styles and lower socioeconomic status, are associated with the link between media use and ADHD characteristics. 

Next, several empirical studies observed links between the *intensity* of social media use and ADHD symptoms. A study that examined adolescents at risk for psychopathology demonstrated that the daily number of text messages sent, as well as daily reported time on digital media, were related to self-reported ADHD (and conduct disorder) characteristics as measured on the same day, and these digital media measures were related to lower self-regulation over an 18-month follow-up period [37]. Other work found that the number of parent- and self-reported social media accounts was positively related to adolescents’ ADHD symptoms [38]. Interestingly, this association was not specific to ADHD symptoms, as the number of social media accounts was also linked to characteristics of oppositional behavior, anxiety, depression, loneliness, and fear of missing out. Finally, a positive link between self-reported distractibility and self-reported amount of time spent on instant messaging platforms was observed in late adolescent college students [39]. Taken together, a higher intensity of social media use seems to be related to higher levels of ADHD, as well as characteristics such as oppositional behavior, anxiety and fear of missing out. 

Other studies demonstrated an association between *problematic* social media use and ADHD characteristics. In a large survey (*n* > 20.000), adolescents reported an association of addictive social media use with hyperactivity, but also with conduct problems and sedentary behavior [40], while another study showed a positive association between self-reported ADHD ratings and self-reported addictive Facebook usage [41]. Moreover, a study that included adolescents with and without ADHD diagnoses reached a similar conclusion: Facebook overuse and addiction, both self-reported, were more frequent in adolescents with ADHD relative to controls [42]. An interesting fact was that adolescents with ADHD also reported more fake Facebook accounts. Besides links with overuse and addiction-like behaviors, ADHD has also been linked to posting more problematic content on social media. A study on 1.3 million tweets from over a thousand Twitter users with self-reported diagnoses of ADHD showed that, relative to matched controls, users with ADHD have significantly different posting behaviors, such that they posted more tweets, did so more often at night time, were more open and used more swear words [43]. Therefore, multiple high quality empirical studies with large samples observed an association between ADHD characteristics and problematic social media use. 

All in all, the evidence for a temporal relationship between social media use and ADHD appears to be mixed and nuanced due to potential confounders, and there are many questions remaining about the mechanisms through which this occurs. Empirical work to date shows most consistent evidence for a relation between problematic social media use and ADHD (symptoms), while intensity of use is linked to ADHD in addition to several other behaviors and outcomes. Consequently, rather than the conclusion that social media potentially may lead to ADHD as a disorder, findings may suggest that they potentially exacerbate or lead to heightened symptoms of inattention, hyperactivity and impulsivity. For future research, it will be important to bridge the distinct literatures from communication sciences and psychiatry to advance our knowledge in this field.

## 3. Integrative Framework

To gain traction on the link between social media use and ADHD, we use neurobiological models of brain development [16,18,21], dual systems and reinforcement sensitivity theories of ADHD [17,19,20,22], combined with theory and empirical work on social development [44,45] to provide an integrative framework (also see Figure 1, left panel). 

### 3.1. Neurobiological Mechanisms

During adolescence, major changes take place in cognitive and social-affective regions of the developing brain [5,46], and it has been postulated that this developmental period is a time of increased social processing [44]. The imbalance model posits that the adolescent brain is especially oriented towards rewards, which can be social (e.g., positive peer feedback) or non-social (e.g., monetary) in nature [16,18,21]. These theories are supported by research showing that brain activity in the reward system (e.g., ventral striatum) peaks in response to reward during adolescence [47,48,49], while brain regions that support cognitive control (e.g., dorsolateral prefrontal cortex) mature at a slower pace. However, the critique regarding the simplicity of these models should be noted; e.g., [50]).

Moreover, neuroimaging research has shown that brain activity in the ventral striatum is dependent on the social context. For example, risky driving in the presence of peers recruits the reward regions of the brain to a greater extent in adolescents, but not (young) adults [51]. A unique fMRI study illustrated that viewing photos on Instagram with many likes, as compared to few likes, was associated with greater activity in reward regions, but also in brain regions associated with social information processing, imitation and attention [52]. In addition, emotional stimuli, such as socially appetitive cues may decrease adolescents’ capacity for inhibitory control through increased activation in affective brain regions such as amygdala and ventral striatum [53]. Taken together, the neural sensitivity to (social) reward may be an important component and reinforcer in the use of social media during adolescence [6]. Nonetheless, it is important to note that much is still unknown about the bidirectional relation between social media use and brain development [54,55].

The dual pathway model suggests two subtypes of ADHD characterized by relatively distinct pathways in the brain, i.e., those with dysregulation of inhibitory control systems and altered reward processing [19,20]. A large body of empirical literature has consistently linked ADHD with hypo-activity in brain regions implicated in cognitive control (ADHD International Consensus statement [56]). Interestingly, some empirical work points to delayed maturation of the brain, while neurodevelopmental theory suggests that disruption of brain development in utero or early life may be associated with neuropsychiatric problems [57,58,59,60]. Theoretical accounts on motivation emphasize differential reward sensitivity and reinforcement learning in ADHD [17,22], but see [61]. In the anticipation of rewards, empirical studies show that the ventral striatum tends to be hypo-active in adolescents with ADHD, yet the findings on the receipt of reward are mixed (reviewed in [62,63]). Thus, the previous work shows that the imbalance between these brain systems, which is often described in typically developing adolescents, is likely larger in those with ADHD [64].

The neurobiological characteristics of ADHD manifest themselves on a behavioral level as impaired executive functions, such as response inhibition, working memory and planning [65,66], and a strong preference for immediate over delayed rewards [67,68,69]. As such, the appeal of digital media for adolescents with ADHD may lie in the fast-paced exchange of information that offers immediate reward and reinforcement [9,10]. Given that the reward systems in the brain show greater activity in a social context [51], we can infer that *social* media may be even more rewarding, which in adolescents with ADHD goes hand in hand with a limited capacity for cognitive control. Thus, from the neurobiological perspective, it is understandable that regular use of social media might turn to problematic use in adolescents with ADHD.

### 3.2. Social Mechanisms: Peer Relationships

Adolescence has been described as a time of ‘social orientation’ [70] during which peer relations are at the top of the priority list, while parents move down the list as adolescents gain their independence [5,46,71]. More time is spent with peers, new social spheres are formed (i.e., larger peer groups), and the quality of friendship changes to become more supportive and intimate [72,73]. Adolescents with ADHD display a host of social difficulties that might contribute to a struggle with these changing social dynamics, and therefore require more complex social skills to navigate adequately (reviewed in [45]). Here, we discuss factors related to peer functioning to understand the link between ADHD and problematic social media use. 

A large body of literature illustrates early social difficulties in children with ADHD, which continue and cascade during adolescence, and may manifest in novel ways [45]. Socially inadequate behaviors such as social intrusiveness, talking when inappropriate, and being easily distracted in conversation [4,74] are associated with impairments in peer relationships and lower popularity among peers [75]. Up to 50–80% of children with ADHD experience peer rejection and victimization [76,77], and peer relationships are qualified by lower rates of stability, quantity and quality in adolescents with ADHD [45]. Within friendships, adolescents with ADHD do not necessarily report more negative interactions, but social support from these friendships decreases over time, while it increases in adolescents without ADHD [78].

One hypothesis posits that due to a lower quality of peer relations and greater experience of peer rejection and victimization in the offline world [79], adolescents with ADHD may be drawn to spending large amounts of time on social media, trying to connect to their peers. Recent evidence showed that only adolescent girls with ADHD, relative to boys with(out) ADHD, and girls without ADHD, had friends whom they initially met online, and fewer of their best friends attended their school [80]. This is in line with research findings in female young adults (mean age 19; with and without childhood ADHD diagnosis), revealing that those with a history of ADHD had a preference for online rather than face-to-face communication, and more often reported online interactions with strangers [81].

Other work notes that social difficulties in adolescents with ADHD in offline and online social situations are generally similar, such that social media is merely a different platform for manifestation of the same problems [82,83,84]. Some of the findings from Mikami et al. [81] are in line with this hypothesis, as childhood ADHD (vs. no childhood ADHD) predicted having fewer Facebook friends and less connection and support within Facebook posts received from friends in young adulthood. Interestingly, face-to-face peer impairment at ages 9.6 and 14.1 mediated these links. Similarly, recent work in an adolescent sample with ADHD (ages 13–16; no control group; 75% boys) demonstrated that offline risks (e.g., poorer social skills, more internalizing symptoms) were associated with online risks (e.g., weak online connections) [82]. Indeed, higher rates of peer rejection may also increase susceptibility to peer influence in adolescents with ADHD, resulting in the display of risk-taking behaviors to gain status or avoid rejection [85,86,87]. The greater propensity for risk-taking, in addition to a lack of appropriate social inhibition, may extend to the online domain in the form of online risk-behaviors such as sexting or cyberbullying [82].

Taken together, the findings reviewed here provide evidence that peer relations are at the top of the online priority list for adolescents with ADHD. Current evidence supports the hypothesis of compensating impaired offline social relations by trying to connect with peers online, albeit especially in girls with ADHD. At the same time findings reveal that online peer problems appear to mimic offline peer problems. Importantly, researchers also note the *connectedness* of the offline and online social world, such that negative online experiences potentially magnify, transform or cause problems in offline social relationships, and vice versa [10,82,83,88]. This seems a key factor to take into account in future research when examining the role of peer relationships related to (problematic) social media use. 

### 3.3. Social Mechanisms: Parental Factors 

While adolescents become increasingly oriented towards peers, parents and the larger family also continue to have a considerable influence on the child’s behaviors (reviewed in [89]). Here, we highlight three parental factors that might play a role in adolescents’ problematic social media use: parental ADHD characteristics, parenting style, and general family circumstances. 

ADHD is a highly heritable condition, which explains why a disproportionate amount of parents of children with ADHD show elevated ADHD characteristics themselves [90]. Parents with high levels of ADHD characteristics are more likely to have problems with their own social media use [43], and may therefore not be an optimal role-model for their children. This is crucial, as earlier work demonstrated that social media use of parents and children are highly correlated [31,91].

Next, children with ADHD often challenge parenting behaviors. Parents of children with ADHD show fewer positive and more negative parenting behaviors and report greater levels of parenting stress (for a systematic review, see [92]). One parenting behavior that is particularly relevant to understand the link between ADHD and social media use is parental monitoring, which is “a set of correlated parenting behaviors involving attention to and tracking of the child’s whereabouts, activities, and adaptations” [93]. Parental monitoring is negatively related to adolescents’ ADHD characteristics [94], and consistently mediates the link between adolescents’ ADHD characteristics and several measures of real-life impairment [95,96]. Similar patterns are likely in the domain of social media. As recently suggested [10], lax parenting and poor monitoring may not provide adolescents with ADHD the guidance they need, and may contribute to higher intensity and more problematic social media use. In typically developing adolescents, suboptimal parenting was linked to an increased motivation to engage in social media activities [97]. Related work shows that parental influences on adolescents’ media behavior can also have opposite effects (i.e., “boomerang-effect”; [98]) when parental involvement is experienced as controlling or inconsistent, whereas effects are positive when parental monitoring was autonomy-supporting [99].

A third and slightly more indirect way to understand the link between ADHD and high intensity/problematic social media use is examining the general family situation. In families of children with ADHD, parent–child relationships are characterized by more negativity relative to families without ADHD [100], and adolescents with ADHD disproportionately often have conflicts with their parents and siblings [101]. Adverse family situations could be linked to (social) media use in several ways. Adolescents could use media to escape from the aversive family climate, they could show positive/non-problematic media use as a response to the negativity in the family context, or they could transfer the negative emotions from the family climate towards the online world. Most evidence points to the latter ‘family context’ hypothesis [102]. Knowing that families of children with ADHD are characterized by increased negativity, combined with evidence for the family context hypothesis, would potentially suggest that beneficial effects of systemic interventions could also impact social media use.

## 4. Discussion: From Neurobiological and Social Mechanisms to Clinical Recommendations for Problematic Social Media Use

In the previous sections, we described literature that consistently demonstrated higher levels of problematic social media use in adolescents with ADHD, and provided an integrative framework that encompasses neurobiological and social mechanisms to explain this association. In this section, we provide recommendations for clinicians on how to target these three mechanisms pertaining to social media use when working with adolescents with ADHD (see Figure 1, right panel) and highlight important research gaps that should be addressed to better understand the phenomenon and ultimately improve the treatment options that are currently available. 

### 4.1. Neurobiology: Pharmacological Interventions 

All major guidelines recommend pharmacological treatment as evidence-based intervention aimed at decreasing ADHD core symptoms [103,104]. Thus far, no studies have directly investigated the effects of stimulant medication on social media use, but more indirect evidence suggests potential beneficial effects. A plausible hypothesis would be that a decrease in core ADHD symptoms would lead to a decrease in the problematic social media use that is associated with these symptoms. Stimulant medication works most prominently in inferior frontal and striatal regions [56], and could theoretically decrease the imbalance between cognitive control and reward systems of the brain, with beneficial effects on problematic social media use. In support of this hypothesis, it was found that school-age children with ADHD and addictive internet use who took methylphenidate for eight weeks, not only decreased in core ADHD characteristics but also in problematic internet use [105]. Future research could extend this work in the adolescent age group and examine whether such effects are also present in the domain of social media use.

### 4.2. Peer Interventions

Peer relations are at the core of social media use, and therefore interventions targeting such relationships could be promising for the treatment of problematic social media use. Generally, two types of peer interventions for ADHD can be distinguished. Social skills training teaches children social skills, which are then supposed to generalize to children’s daily lives. However, empirical evidence shows that these interventions are not very effective [106]. Other behavioral peer interventions either involve parents (e.g., Parent Friendship Coaching [107,108]) or staff working with children with ADHD in recreational settings (e.g., summer treatment [109]) and aim to improve social functioning of children by intervening in the setting in which the behavior change is desired. These treatments are considered well-established for ADHD [106,110], with pronounced improvements in social functioning. 

Given that social media behaviors either mimic real-life social problems and/or are compensatory for these real-life social problems, it is expected that behavioral peer interventions would also have a positive impact on social media use. This hypothesis, however, awaits empirical testing. An important caveat is that all peer interventions for ADHD are investigated in school-age children, whereas problematic social media use becomes most prominent in adolescence. Although improving peer functioning in childhood is likely to prevent a cascade of problematic social functioning that continues into adolescence, there seems an urgent need to develop interventions that tackle social dysfunctioning, including problematic social media use, in adolescents specifically. 

### 4.3. Parenting Interventions

For school-age children with ADHD, behavioral parent training is recommended as first line evidence-based intervention [111,112]. As children with ADHD are at risk to develop problematic social media use, and parenting is a relevant mechanism in this link, parenting interventions in childhood could potentially prevent problematic social media use in adolescence. 

First, parents are the most proximal role models of their children, and their own behavior with regard to social media is therefore important to consider. The amount of parental distraction with technology (so-called technoference) is crucial, as this predicts internalizing as well as externalizing problem behaviors in their children [113]. A qualitative study of mobile device use during parent–child dinner in restaurants showed several response styles for children in case of parental technoference: some of the children became passive as the conversation decreased, whereas others displayed more misbehavior in order to obtain their parents’ attention. In response, parents with high levels of absorption towards their mobile devices responded more harshly to problem behavior of their children [114]. Emphasis on modeling of healthy social media behavior may be particularly relevant for parents of children with ADHD, as it is plausible that many of these parents have ADHD-related problematic social media use themselves, given the genetic nature of ADHD [90].

Second, parenting interventions may focus on improving parenting style, given the association between lax parenting and problematic social media use [10]. Research on parental knowledge shows that adolescents’ voluntary disclosure of information about their activities predicts decreases in risky behavior (for a review, see [99]), whereas harsh monitoring and involuntary surveillance, which also lead to increased parental monitoring, have lower or even iatrogenic effects [98,99]. As behavioral parent training for children with ADHD usually leads to increases in positive parenting and the parent–child relationship and decreases in negative parenting (for a meta-analysis, see [112,115]), this intervention is likely to contribute to an atmosphere in which children are more willing to voluntarily involve their parents in their social media use. 

Behavioral parent training could also target problematic social media use directly, although evidence for this intervention almost exclusively comes from studies on school-age children. A recent randomized controlled micro-trial on brief personalized parent training showed that several behavioral therapeutic techniques (i.e., stimulus-control techniques, contingency management) were effective in reducing problem behaviors that were selected by parents [116]. This demonstrates that behavioral parent training could be easily personalized to meet the needs of specific families, and could therefore also target problematic social media use, in case this is relevant. Taking a social-ecological approach, it may be helpful for some parents to examine the role of digital media in family dynamics more broadly [117]. Other parents may be helped with recommendations on how to cope with problematic social media use, specifically for their child with ADHD. Helpful resources for general guidelines around digital media usage in children and youth are provided by the American Academy of Pediatrics [118] and American Academy of Child and Adolescent Psychiatry [119]. These recommendations could involve restricting social media use or monitoring both the content and the circumstances of social media use (for more detailed recommendations in adolescents with ADHD, see [120]). 

## 5. Future Directions to Improve the Treatment of Problematic Social Media Use in Adolescents with ADHD

In the next section, we highlight two related avenues in treatment and research that may be useful future directions in the treatment of problematic social media use in adolescents with ADHD (symptoms). First, characteristics of problematic social media use have much in common with addiction (i.e., preoccupation, inability to stop, neglect of health, negative impact on life functioning), and interventions that are beneficial to combat other forms of addiction (e.g., substance abuse, internet/gaming addiction) could also be promising to target problematic social media use. Second, sleep problems and (social) media use are heavily intertwined, and interventions targeting one of those aspects potentially also impact the other. Although the link between social media and sleep is highly intuitive, there is currently a scarce body of empirical work examining this relation. 

### 5.1. Cognitive Behavioral Therapy

Indirect indications for the effectiveness of cognitive behavioral therapy (CBT) as intervention for problematic social media use comes from literature on internet/gaming “addiction” [121,122,123]. CBT could target problematic social media use in a more direct way, although CBT aimed at social media use has not yet been investigated for adolescents with ADHD. As a first step, the function of problematic social media use should be established. Several different functions for social media use have been suggested, such as avoidance/escapism (of loneliness, of troubled real-life relationships with parents/peers), boredom/passing time, companionship, emotion regulation/negative affect management, entertainment, relationship maintenance, and socialization [42,121]. Behavioral experiments can be conducted to find more beneficial ways of coping and simultaneously, cognitive biases underlying the maladaptive social media behaviors could be identified (e.g., “I’m never able to have any meaningful offline friendships”). Group CBT may be favorable over individual CBT, as it fosters offline social skills, increases engagement and often a support network is established within the group [121].

### 5.2. Sleep

Media use in general is associated with sleep problems in children and adolescents, as more screen time is linked to adverse sleep outcomes [124]. Key mechanisms to understand this link are the displacement of time (screen time instead of sleep time) and effects of the light of the devices on the circadian rhythm [125]. Both are relevant in the context of ADHD, as ADHD is consistently associated with later sleep onset and shorter sleep duration, more daytime sleepiness and more disturbances in sleep [126,127], and disturbances in circadian rhythm are disproportionately often reported by and observed in adolescents with ADHD [127]. However, the number of studies directly addressing the interplay between ADHD, social media use, and sleep problems is remarkably scarce.

A study investigating adolescents with ADHD found that, on average, they spent 5.31 h on nighttime media use (i.e., the use of digital media after 9 pm; potentially using multiple digital media at the same time) [128]. Moreover, after controlling for potential confounders as medication and ADHD severity, nighttime media use was linked to elevated self- and parent-reported sleep problems, shortened sleep duration, sleepiness during the day and self-reported internalizing problems. Another study including adolescents with and without ADHD found that adolescents with ADHD made more use of phone and video chatting, and found that this technology use was linked to adverse sleep outcomes, while controlling for relevant confounders [129]. Finally, a systematic review on sleep problems in adolescents with ADHD concluded that reduced parental monitoring—which occurs disproportionately often in adolescents with ADHD—on evening activities such as social media use was linked with later bed-time, more daytime sleepiness and decreased overall sleep time [130].

The paucity of studies on ADHD, social media use and sleep, as well as the correlational nature of the existing body of work, leave many open questions. First, the studies reviewed above tentatively suggest that excessive social media use could cause sleep problems. The opposite direction is also likely, as sleep problems may cause executive dysfunctioning [131], which may in turn lead to lower impulse control regarding the temptations of social media. Given this putative bidirectionality, intervention studies on sleep problems are advised to include (social) media use as an outcome measure, and also as a potential mechanism of change. Further research on this topic is highly vital in order to inform clinical recommendations in the future.

## 6. Strengths and Limitations

This review has several strengths, including an integrative framework that encompasses neurobiological and social components, the connection of theories from research in adolescence and research on ADHD with clinical research, as well as a hands-on list of clinical recommendations. However, several limitations warrant mentioning. First, the current review is a narrative review and our search process was not entirely systematic. However, we do believe that we have included the most recent and relevant findings on ADHD and problematic social media use in adolescence. Second, most of the clinical recommendations are based on indirect associations, as there are hardly any intervention studies directly measuring their impact on problematic social media use. Future research could build on our work and design interventions based on the described mechanisms that target problematic social media use more directly, in order to examine the effects in children and adolescents with ADHD. 

## 7. Conclusions

The evidence summarized in this review shows that adolescents with ADHD (symptoms) are likely to display a high intensity and problematic use of social media. From a neurobiological perspective, we explain the appeal of social media from an imbalance between the development of brain regions involved in reward together with limited cognitive control, especially in the social domain. In the social world of adolescents, we note the connectedness of the offline and online social world of peer relationships, such that negative experiences in one social domain may magnify or change experiences in the other. In the family situation, parents may impact social media use by setting an example with own use, employing parental monitoring techniques and by the transfer of emotions from family conflicts to social media. In our clinical recommendations, we illustrate how existing evidence-based interventions can be utilized to improve social media use, and highlight the need of future research into interventions that may specifically target social media use in adolescents with ADHD.

## Figures and Tables

**Figure 1 brainsci-12-01625-f001:**
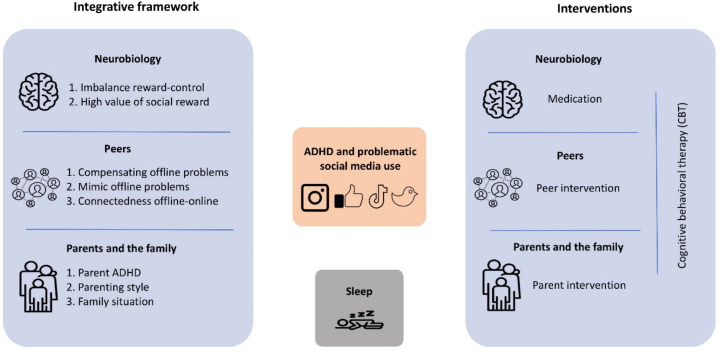
Integrative framework of mechanisms involved in problematic social media use in adolescents with ADHD (left panel) and potential interventions (right panel).

## Data Availability

Not applicable.

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
