# Peer review of "Understanding Problematic Social Media Use in Adolescents with Attention-Deficit/Hyperactivity Disorder (ADHD): A Narrative Review and Clinical Recommendations"

_brainsci, 2022, doi:10.3390/brainsci12121625_

Round 1
Reviewer 1 Report
10 November 2022
Manuscript ID: brainsci-2044776
Type: Article
Title: ‘Adolescent Problematic Social Media Use and Attention-Deficit/Hyperactivity Disorder (ADHD): A Brief Review and Clinical Recommendations’ by Dekkers TJ and van Hoorn J, submitted to Brain Sciences
Dear Authors,
The present review article by Dekkers and van Hoorn, entitled ‘Adolescent Problematic Social Media Use and Attention-Deficit/Hyperactivity Disorder (ADHD): A Brief Review and Clinical Recommendations’ is a well-written and useful summary on the status of knowledge of the problematic use of social media in adolescents with ADHD. For this purpose, authors selected recent studies that focused on neurobiological characteristics of ADHD and on social mechanisms in adolescents with this disorder, to identify how different interventions may work on the specified mechanisms and provide clinical recommendations.
In general, I think the idea of this review is really interesting and the authors’ fascinating observations on this timely topic may be of interest to the readers of Brain Sciences. However, some comments, as well as some crucial evidence that should be included to support the author’s argumentation, needed to be addressed to improve the quality of the manuscript, its adequacy, and its readability prior to the publication in the present form, in particular reshaping parts of the Introduction and Discussion sections by adding more evidence and theoretical constructs.
Please consider the following comments:
1. Title: I would suggest modifying the title that, in my opinion, is too long and appear to be unfocused and not so representative of the contents of the article.
2. Abstract: According to the Journal’s guidelines, please expand the abstract to 200 words, proportionally presenting the background, the objectives, the summary, and the conclusion. The background should starts with general one, then detailed one and the current issue addressed to this study and the conclusion should address the potential and the advance this review has provided in the field.
3. Keywords: Please list ten keywords and use them as many as possible in the first two sentences of the abstract.
4. A graphical abstract is highly recommended.
5. Please declare the type of review, narrative, scoping, or systematic one and this should be done in the abstract. I would ask the Authors to clarify the criteria they decided to use for studies’ collection in their review: they should specify the number of studies included in the review and the requirements used to decide whether a study met the inclusion/exclusion criteria of the review; they also should provide a more detailed description of all other variables for which data were sought, and briefly present results of all statistical syntheses conducted.
6. Neurobiological mechanisms: The ‘Introduction’ section is well-written and nicely presented, with a good balance of descriptive text and information about characteristics and phenotypic expressions of ADHD. Even though the authors decided to take a narrow view of mechanisms underlying relation between ADHD and social media use, I believe that a deeper examination of the neurobiological basis of this developmental disorder, with a specific focus on how ADHD is associated with neurobehavioral reward system dysfunctions that pose debilitating impairments in adaptive decision-making and inhibitory control, would provide a useful background. Interestingly, results from recent studies outlined typical dysfunctional behaviors in patients with ADHD, such as deficit in action control, rapid vigilance and motor inhibition (https://doi.org/10.3389/fnbeh.2022.998714; https://doi.org/10.1007/s00702-022-02513-5), which are associated with psychopathological conditions, which are characterized by severe impulsivity problems that can determine significant impairment or distress (due to poor regulation and capacity of control, which can be intensified in the presence of emotional stimuli) (https://doi.org/10.3389/fnbeh.2022.946263; https://doi.org/10.3390/cells11162607).
7. I would ask the Authors to include a Discussion section, to help readers understand the main findings and implications of the review. In this final section, authors could describe the results of their review and their argumentation and capture the state of the art, trying to explain the theoretical implication as well as the translational application of this paper, to adequately convey what they believe is the take-home message of their study. In this regard, I believe that it would be necessary to discuss theoretical and methodological avenues in need of refinement, as well as suggestions of a path forward in understanding the possibility of target tailored interventions to better understand social media use in adolescents with ADHD.
8. In according to the previous comment, I would ask the authors to also include a proper ‘Limitations and future directions’ section before the end of the manuscript, in which authors can describe in detail and report all the technical issues brought to the surface.
9. Figure: According to the Journal’s guidelines, please add an explanatory caption for each figure/table within the text.
10. References: The authors should consider revising the bibliography, as there are several incorrect citations. Indeed, according to the Journal’s guidelines, they should provide the abbreviated journal name in italics, the year of publication in bold, the volume number in italics for all the references. Also, please correct in-text citations: reference should be numbered, and placed in square brackets [ ] (for example [1]).
11. I suggest submitting your work to an English native speaker to help with some grammar mistakes that can be found in different sections of the manuscript.
Overall, I believe that the manuscript might carry important value describing recent evidence on the relation between ADHD and social media use, providing an integrative framework of explanatory mechanisms for social media use from a neurobiological perspective as well as social perspective. I hope that, after these careful revisions, this paper can meet the Journal’s high standards for publication. I am available for a new round of revision of this paper.
I declare no conflict of interest regarding this manuscript.
Best regards,
Reviewer
Author Response
see attached document

Reviewer 2 Report
This paper is very good, and tackles a very real and challenging issue that parents and clinicians deal with daily. There are limited studies and data, and the authors summarize studies, put the data/issues into a theoretical framework and make recommendations about potential treatments. Specific comments:
- on line 43 - the reference for Valkenburg, Meier, & Beyens 2022 is not listed in the References section at the end of the paper
- On line 110 - the abbreviation CD is used for the first time, without using the full term prior to this. It should be written out in full the first time a term is used ie Conduct Disorder (CD).
- Line 198 - I believe the authors intended to write: "... developing adolescents in likely larger in than those with ADHD."
- In section 3.2, the authors are very clear that the effective interventions for peer relations are in school aged children, and there are not data for adolescents; in section 3.3, when discussing parenting interventions, the authors start by discussing the fact that the parenting interventions with evidence are in school aged children. I feel that this point is lost as this section continues; as discussions of the 'microtrial' (lines 399-403) do not clearly state that the study was in school aged children (up to 12 years old), and without clearly emphasizing this point, I believe the reader is led to believe that these interventions have evidence for parents of adolescents, which is not the case. I suggest more clearly stating the fact that these interventions have evidence that they help in childhood; and theoretically they may help in adolescence.
- One of the strengths of this paper, in my review, is that the authors pull together ideas and theories to help us understand a rapidly growing and changing problem - when we have limited data and research. That said, there are other papers published which may provide guidance.
- The authors reference Ceranoglu 2018 on line 409 - and suggest reviewing the list of recommendations for digital media use for parents of children and youth with ADHD. I have mixed feelings about this approach. On the one hand, this sends the reader to another reference which has more specific guidance, on the other hand, the authors shouldn't repeat all the recommendations that Ceranoglu has published. Perhaps the authors could include a summary of the recommendations? or some of the main points? Furthermore, to my point above, the research referenced about parenting interventions in ADHD are based on children (not teens), where Ceranoglu makes recommendations around interventions/supports in children and youth. There is also reference made to the recommendations of the American Academy of Pediatrics and American Academy of Child and Adolescent Psychiatry around guidance for digital media usage. While these are not empirically studied, they are published by major organizations. The authors of this paper do not discuss/acknowledge these (potentially because they are not specific to ADHD).
- Furthermore, Dalope et al 2018 (https://doi.org/10.1016/j.chc.2017.11.001) used different frameworks to consider understanding problematic digital media usage in children and youth. The authors discuss the social-ecological model, the family systems theory, and a developmental framework. They also reference the recommendations from the American Academy of Pediatrics around digital media use recommendations. While this paper does not address ADHD specifically, it contains frameworks and interventions which may be relevant. It seems the authors have chosen not to include the social-ecological model - which may be relevant. I wonder if the authors have considered this as part of the model? or if there was a particular reason it was not included in their framework/model. In some cases, not only could the social-ecological model be relevant to understand ADHD, it could also lead to interventions/programs to assist youth in certain communities/lower socioeconomic areas (though I have not searched for published papers on this in adolescents with ADHD).
Round 2
Reviewer 1 Report
23 November 2022
Manuscript ID: brainsci-2044776
Type: Article
Title: ‘Adolescent Problematic Social Media Use and Attention-Deficit/Hyperactivity Disorder (ADHD): A Brief Review and Clinical Recommendations’ by Dekkers TJ and van Hoorn J, submitted to Brain Sciences
Dear Authors,
I am pleased to see that the authors took my comments seriously and solved most of the issue I raised in the previous round of the peer-review session. Currently, the manuscript is a well written and nicely presented review paper highlighting the status of knowledge of the problematic use of social media in adolescents with attention deficit hyperactivity disorder (ADHD). That said, I just suggest a couple of suggestions, I believe, which help the authors improve the quality of the manuscript to finalize my part of the peer review session.
1. The figure perfectly serves as a graphical abstract. Please try to place a graphical abstract because it helps readers to digest and understand the core tip of the review, leading to more citation of this paper to be.
2. The introduction may help a bit more information on the neurobiological background of ADHD. Those papers may help the authors inspire this aspect (doi: 10.3371/CSRP.NCLQ.053122; https://doi.org/10.3389/fnbeh.2022.998714; https://doi.org/10.3390/cells11162607).
3. References: Generally, they are corrected. Please place a period in the end of journal’s abbreviations and doi number should be placed like “doi:” without “http:”.
After these careful revisions, this paper can meet the Journal’s high standards for publication. I am looking forward to seeing further works by the authors in near future.
Thank you for your work.
I declare no conflict of interest regarding this manuscript.
Best regards,
Reviewer